# The Query Complexity of Cake Cutting

**Simina Brânzei**
Purdue University, USA
simina.branzei@gmail.com

**Noam Nisan**
Hebrew University of Jerusalem, Israel
noam@cs.huji.ac.il

## Abstract

We consider the query complexity of cake cutting in the standard query model and give lower and upper bounds for computing approximately envy-free, perfect, and equitable allocations with the minimum number of cuts. The lower bounds are tight for computing contiguous envy-free allocations among $n = 3$ players and for computing perfect and equitable allocations with minimum number of cuts between $n = 2$ players. For $\epsilon$-envy-free allocations with contiguous pieces, we also give an upper bound of $O(n/\epsilon)$ and lower bound of $\Omega(\log(1/\epsilon))$ queries for any number $n \geq 3$ of players.

We also formalize moving knife procedures and show that a large subclass of this family, which captures all the known moving knife procedures, can be simulated efficiently with arbitrarily small error in the Robertson-Webb query model.

## 1 Introduction

We consider the classical cake cutting problem due to Steinhaus [62], which captures the fair division of a heterogeneous resource—such as land, time, mineral deposits, fossil fuels, and clean water [56]—among several parties with equal rights but different interests over the resource. This model has inspired a rich body of literature in mathematics, political science, economics [58, 11, 49], and computer science [15, 27]. Implementations of fair division algorithms can be found on Spliddit [45].

Mathematically, the cake is represented as the unit interval $[0, 1]$. There is a set of $n$ players with valuation functions induced by probability measures over $[0, 1]$. Given such a resource, the goal is to compute an allocation of the cake in which every player is content with the piece received. A major challenge for a mediator trying to compute a desirable allocation is that the preferences of the players are private. Discrete cake cutting protocols operate in a query model, due to Robertson and Webb [68], in which a center that does not know the players asks them *cut* and *evaluate* queries until it manages to extract enough information about their preferences to determine a fair allocation.

An example of a cake cutting protocol is Cut-and-Choose, which dates back to more than 2500 years ago when it appeared in written records in the context of land division. Cut-and-Choose can be used to obtain an envy-free allocation among two players, Alice and Bob, trying to divide a heterogeneous resource (e.g. a birthday cake with different toppings): First Alice cuts the cake in two pieces of equal value to her, then Bob picks his favorite and Alice takes the remainder.

**Fairness Notions.** Two of the prominent fairness notions, envy-freeness and proportionality, have been the subject of in depth study from a computational point of view. Proportionality requires that each player gets their fair share of the resources, which is their total value for the whole cake divided by the number of players, while envy-freeness is based on social comparison and means no player should want to swap their piece with anyone else's. The two notions are not necessarily comparable, since envy-freeness can be trivially achieved by throwing away the entire resource, which is not true for proportionality. However, when the entire cake must be allocated, envy-freeness implies proportionality and can be surprisingly hard to reach.

36th Conference on Neural Information Processing Systems (NeurIPS 2022).

The problem of finding an envy-free cake cutting protocol was suggested by Gamow and Stern [41] and solved by Selfridge and Conway for three players (cca. 1960, see, e.g., [58, 11]) and by Brams and Taylor [14] for any number of players. From a computational point of view, the Brams and Taylor protocol has the major drawback that its runtime can be made arbitrarily long by setting up the valuations of the players appropriately. In 2016, Aziz and Mackenzie [6] announced a breakthrough by giving the first bounded envy-free cake cutting protocol for any number of players, where bounded means that the number of queries is only a function of the number of players, and not of the valuations.

The query complexity of proportionality is well understood. The problem of computing a proportional allocation with contiguous pieces can be solved with $O(n \log n)$ queries in the Robertson-Webb model by running a protocol due to Even and Paz [36], with a matching lower bound due to Woeginger and Sgall [68] for contiguous pieces; this was extended to arbitrary allocations by Edmonds-Pruhs [34].

In contrast, for the query complexity of envy-free cake cutting a lower bound of $\Omega(n^2)$ was given by Procaccia [55] and an upper bound of $O\left(n^{n^{n^{n^{n}}}}\right)$ by Aziz and Mackenzie [6]. The algorithm in [6] outputs highly fragmented allocations on some instances. Envy-free allocations with contiguous pieces do in fact exist for very general valuations [1] (see, e.g., [61, 63, 65]), but cannot be computed in the Robertson-Webb model [64] except for specific classes such as polynomial functions [16].

**Approximate Fairness.** In light of this impossibility, it makes sense to study the computation of $\epsilon$-envy-free allocations with few cuts. For a general utility model where the valuations are arbitrary functions (not necessarily induced by probability measures), bounds on the query complexity of approximate envy-free cake cutting with contiguous pieces were given in [32] together with a proof of PPAD-hardness. While the problem of computing contiguous $\epsilon$-envy-free allocations is in PPAD [65], neither the computational hardness nor the query lower and upper bounds in [32] carry over (or are close to tight) in the usual cake cutting model where the valuation functions are induced by probability measures, which leaves wide open the complexity of this problem.

Aside from proportionality and envy-freeness, a third notion of fairness is known as equitability, in which each player must receive a piece worth the same value. Equitable and proportional allocations with contiguous pieces were shown to exist by Cechlarova, Dobos, and Pillarova [24]. The computational complexity of approximate equitability was investigated by Cechlarova and Pillarova [25], who gave an upper bound of $O\left(n\left(\log n + \log \epsilon^{-1}\right)\right)$ for computing an $\epsilon$-equitable and proportional allocation with contiguous pieces for any number of players, and by Procaccia and Wang [57] who showed a lower bound of $\Omega\left(\log \epsilon^{-1}/\log \log \epsilon^{-1}\right)$ for finding an $\epsilon$-equitable allocation (not necessarily contiguous) for any number of players.

More stringent fairness requirements are also possible, such as $(i)$ the necklace splitting problem, for which the existence of fair solutions was established by Neyman [50], with a bound on the number of cuts given by Alon [2], $(ii)$ the more general notion of exact division [2], which generalizes necklace splitting and was shown to exist by [33], and $(iii)$ the competitive equilibrium from equal incomes, the existence of which was determined by Weller [67]. The necklace splitting problem is contained in PPA and in fact is PPAD-hard [38]. A well known instantiation of the necklace splitting solution is that of perfect allocations, which are simultaneously proportional, envy-free, and equitable.

**Related Work.** The complexity and existence of various fairness concepts in cake cutting and related models, such as cake cutting where the resource is a chore, pie, multiple homogeneous goods, multiple discrete goods have also been studied (see, e.g., [29, 52, 4, 22, 42, 30, 17, 13, 23, 53, 9, 1, 18]). The communication complexity of cake cutting was studied in [21].

Since the initial manuscript of our paper was circulated, there have been many follow-up works. The complexity of cake cutting (e.g. envy-free division, consensus halving, necklace splitting) was studied, e.g., in [31, 43, 28, 44, 40, 39, 3, 54, 8]. Indivisible goods were studied, e.g., in [51] for their query complexity and in [48, 26] for algorithms. Cake cutting with separation was studied in [35], fair division of a graph in [10], multi-layered cakes in [46], fair cutting in practice in [47], and cake cutting where some parts are good and others bad in [59] and when the whole cake is a "bad" in [37].

---

[1]Such allocations exist even for valuations not induced by probability measures (see, e.g., [61], [63] for a proof based on a topological lemma on intersection of sets, and [65] for a proof using Sperner's lemma).

[2]The problem of exact division is the following: given a cake with $n$ players and target non-negative weights $w_1 \ldots w_k$, find a partition $A = (A_1, \ldots, A_k)$ such that $V_i(A_j) = w_j$ for each player $i$ and every piece $A_j$.

# 2 Our Contribution

We consider the query complexity of cake cutting in the standard Robertson-Webb (RW) query model [68] for several fairness notions: envy-free, perfect, and equitable allocations with the minimum number of cuts. Such allocations are known to exist for each instance, but several impossibility results preclude their computation in the standard query model [64, 25].

Nevertheless, the computation of approximately fair solutions is possible by discretizing the cake in very small pieces. For the fairness notions we consider, a number of queries that is polynomial in $n$ and $1/\epsilon$ suffices to find $\epsilon$-fair allocations for several fairness concepts, such as $O(n/\epsilon)$ for contiguous envy-free allocations and $O(n(k-1)/\epsilon)$ for $(\epsilon, k)$-measure splittings.

## 2.1 Lower Bounds

Our first main contribution is to give lower bounds for the problems of computing approximately fair allocations (with deterministic protocols).

**Theorem 2.1.** *Consider a cake cutting problem. In the standard (RW) query model, for each $\epsilon > 0$:*

- *Finding an $\epsilon$-envy-free allocation with contiguous pieces for $n \geq 3$ players, where $n$ is fixed, requires $\Omega\left(\log \frac{1}{\epsilon}\right)$ queries.*

- *Finding an $\epsilon$-equitable allocation with contiguous pieces for $n = 2$ players requires $\Omega\left(\log \frac{1}{\epsilon}\right)$ queries.*

- *Finding an $\epsilon$-perfect allocation with two cuts for $n = 2$ players requires $\Omega\left(\log \frac{1}{\epsilon}\right)$ queries.*

In the case of perfect allocations for two players, we are always guaranteed to have a solution with at most two cuts, while a solution with one cut may not exist. The lower bounds for envy-freeness and perfect allocations are the first query lower bounds for these problems. The main idea underpinning these results is that of maintaining a self-reducible structure throughout the execution of a protocol, which may be useful more generally for obtaining other lower bounds.

## 2.2 Upper Bounds

We give the following upper bounds for these fairness notions.

**Theorem 2.2.** *Consider a cake cutting problem. In the standard (RW) query model, for each $\epsilon > 0$:*

- *An $\epsilon$-envy-free allocation with contiguous pieces for $n = 3$ players can be found with $O(\log \frac{1}{\epsilon})$ queries and for $n \geq 4$ players with $O(n/\epsilon)$ queries.*

- *An $\epsilon$-perfect allocation with at most two cuts for $n = 2$ players can be found with $O(\log \frac{1}{\epsilon})$ queries.*

The upper bounds do not assume any restrictions on the valuations and the proofs rely on approximately simulating in the RW model two moving knife procedures, due to Barbanel-Brams [7] and Austin [5], respectively. An upper bound of $O\left(\log \frac{1}{\epsilon}\right)$ for computing an $\epsilon$-equitable allocation with contiguous pieces between two players was given by Cechlarova and Pillarova [25].

A summary of our bounds for the query complexity of computing fair allocations can be found in Table 1, together with the results from previous literature.

## 2.3 Moving Knife Protocols

Our second main contribution is to formalize the class of moving knife procedures, which was previously viewed as disjoint from the RW query model. Using this definition, we show that any fair moving knife procedure with a fixed number of players and devices can be simulated in $O(\log \frac{1}{\epsilon})$ queries in the RW model when the players have value densities that are bounded from above by a constant. The next theorem is written for an abstract fairness notion, which we later define.

**Theorem 2.3.** *Consider a cake cutting problem where the value densities are bounded from above and below by strictly positive constants. Let $\mathcal{M}$ be an RW moving knife protocol with at most $r$ steps, such that $\mathcal{M}$ outputs $\mathcal{F}$-fair allocations demarcated by at most a constant number $C$ of cuts.*

*Then for each $\epsilon > 0$, there is an RW protocol $\mathcal{M}_\epsilon$ that uses $O\left(\log\frac{1}{\epsilon}\right)$ queries and computes $\epsilon$-$\mathcal{F}$-fair partitions demarcated with at most $C$ cuts.*

| Fairness notion | Players | Upper bound | Lower bound |
|---|---|---|---|
| $\epsilon$-envy-free (connected) | $n = 2$ | $1$ | $1$ |
| | $n = 3$ | $O(\log\epsilon^{-1})$ $(*)$ | $\Omega(\log\epsilon^{-1})$ $(*)$ |
| | $n \geq 4$ | $O(n/\epsilon)$ $(*)$ | $\Omega\left(\log\epsilon^{-1}\right)$ $(*)$ |
| $\epsilon$-perfect (min cuts) | $n = 2$ | $O(\log\epsilon^{-1})$ $(*)$ | $\Omega(\log\epsilon^{-1})$ $(*)$ |
| | $n \geq 3$ | $O\left(n^3/\epsilon\right)$ [19] | $\Omega\left(\frac{\log\epsilon^{-1}}{\log\log\epsilon^{-1}}\right)$ [57] |
| $\epsilon$-equitable (connected) | $n = 2$ | $O(\log\epsilon^{-1})$ [25] | $\Omega(\log\epsilon^{-1})$ $(*)$ |
| | $n \geq 3$ | $O\left(n\left(\log n + \log\epsilon^{-1}\right)\right)$ [25] | $\Omega\left(\frac{\log\epsilon^{-1}}{\log\log\epsilon^{-1}}\right)$ [57] |
| envy-free (exact) | $n \geq 2$ | $O\left(n^{n^{n^{n^{n^n}}}}\right)$ [6] | $\Omega(n^2)$ [55] |
| proportional (exact) | $n \geq 2$ | $O\left(n\log n\right)$ [36] | $\Omega(n\log n)$ [68, 34] |

Figure 1: Query complexity of cake cutting in the standard query model. Our results are marked with $(*)$. The lower bounds for finding $\epsilon$-perfect and $\epsilon$-equitable allocations for $n \geq 3$ players hold for any number of cuts [57]. The bounds for exact envy-free and proportional allocations hold for any number of cuts, except the upper bound for proportional works for contiguous pieces.

This simulation immediately implies that all the known moving knife procedures, such as the procedures designed by Austin, Barbanel-Brams, Webb [66], and Brams-Taylor-Zwicker [12], can be simulated efficiently within $\epsilon$-error in the RW model when the measures of the players are bounded. In this context we also found a moving knife procedure for computing contiguous equitable allocations for any number of players, which was discovered independently by Segal-Halevi [60]. Note that our version of the protocol works only for hungry valuations (i.e. when the value densities are strictly positive everywhere), while the protocol in [60] works for value densities that could be zero as well.

## 3 Model

The resource (cake) is represented as the interval $[0, 1]$. There is a set of players $N = \{1, \ldots, n\}$ interested in the cake, such that each player $i \in N$ is endowed with a private *valuation function* $V_i$ that assigns a value to every subinterval of $[0, 1]$. These values are induced by a non-negative integrable *value density function* $v_i$, so that for an interval $I$, $V_i(I) = \int_{x \in I} v_i(x)\, dx$. The valuations are additive, so $V_i\left(\bigcup_{j=1}^m I_j\right) = \sum_{j=1}^m V_i(I_j)$ for any disjoint intervals $I_1, \ldots, I_m \subseteq [0, 1]$. The value densities are non-atomic, and in fact sets of measure zero are worth zero to a player. Without loss of generality, the valuations are normalized to $V_i([0, 1]) = 1$, for all $i = 1 \ldots n$.

A *piece of cake* is a finite union of disjoint intervals. A piece is *contiguous* (or connected) if it consists of a single interval. An *allocation* $A = (A_1, \ldots, A_n)$ is a partition of the cake among the players, such that each player $i$ receives the piece $A_i$, the pieces are disjoint, and $\bigcup_{i \in N} A_i = [0, 1]$.

We say that a value density $v$ is: $(i)$ *hungry* if $v(x) > 0$ for all $x \in [0, 1]$, $(ii)$ *uniform* on an interval $[a, b]$ if $v(x) = 1$ for all $x \in [a, b]$, and $(iii)$ *bounded* on interval $[a, b]$ if there exists a constant $D > 0$ such that $v(x) \geq D$ for all $x \in [a, b]$ (or simply bounded if $a = 0$ and $b = 1$). We sometimes to refer to a valuation (or player) as hungry to mean that the corresponding density is strictly positive.

### 3.1 Fairness Notions

An allocation $A$ is said to be *proportional* if $V_i(A_i) \geq 1/n$ for all $i \in N$; *envy-free* if $V_i(A_i) \geq V_i(A_j)$ for all $i, j \in N$; *perfect* if $V_i(A_j) = 1/n$ for all $i, j \in N$; *equitable* if $V_i(A_i) = c$, for all $i \in N$ and some value $c \in [0, 1]$. In addition to allocations, we will refer to partitions as divisions of the cake into pieces where the number of pieces need not equal the number of players. A partition $A = (A_1, \ldots, A_k)$ is a $k$-measure splitting if $V_i(A_j) = 1/k$ for each player $i$ and piece $A_j$.

Envy-free allocations with contiguous pieces and equitable allocations with contiguous pieces always exist [65, 24], while $k$-measure splittings exist with $n(k-1)$ cuts [2].

We will also be interested in $\epsilon$-fair division, where the fairness constraints are satisfied within $\epsilon$-error; for instance, an allocation $A$ is $\epsilon$-envy-free if $V_i(A_i) \geq V_i(A_j) - \epsilon$, for each $i, j = 1 \ldots n$, and an $(\epsilon, k)$ measure splitting if $V_i(A_j) \in (1/k - \epsilon, 1/k + \epsilon)$ for each $i = 1 \ldots n, j = 1 \ldots k$.

We consider an abstract definition for fairness notions that admit approximate versions as follows.

**Definition 1** (Abstract fairness). *A fairness notion $\mathcal{F}$ must satisfy the following condition:*

- *Let $A = (A_1, \ldots, A_n)$ be an allocation that is $\mathcal{F}$-fair with respect to valuations $\vec{v} = (v_1 \ldots v_n)$. For each $\epsilon > 0$ and valuations $\vec{v}' = (v_1' \ldots v_n')$, if $|V_i(A_j) - V_i'(A_j)| \leq \epsilon/2$, $\forall i, j \in N$, then allocation $A$ is $\epsilon$-$\mathcal{F}$-fair with respect to valuations $\vec{v}'$.*

Proportionality, envy-freeness, and perfection are instantiations of this notion of abstract fairness. E.g., if an allocation $A$ is envy-free with respect to valuations $\vec{v}$, then $A$ is also $\epsilon$-envy-free for valuations $\vec{v}'$ that are close to $\vec{v}$.

### 3.2 Query Complexity

All the discrete cake cutting protocols operate in a query model known as the Robertson-Webb model (see, e.g., the book [58]), which was explicitly stated in [68]. In this model, the protocol communicates with the players using the following types of queries:

- ***Cut$_i(\alpha)$***: Player $i$ cuts the cake at a point $y$ where $V_i([0, y]) = \alpha$, where $\alpha \in [0, 1]$ is chosen arbitrarily by the mediator [3]. The point $y$ becomes a *cut point*.
- ***Eval$_i(y)$***: Player $i$ returns $V_i([0, y])$, where $y$ is a previously made cut point.

An RW protocol asks the players a sequence of cut and evaluate queries, at the end of which it outputs an allocation demarcated by cut points from its execution (i.e. cuts discovered through queries). Note that the value of a piece $[x, y]$ can be determined with two Eval queries, $Eval_i(x)$ and $Eval_i(y)$.

A second class of protocols is known as "moving-knife" (or continuous) procedures, which typically involve sliding multiple knives across the cake, while evaluating the players' valuations until some stopping condition is met. This class has not been formalized until now. Examples of moving knife procedures include Austin's procedure, which computes a perfect allocation for two players, Stromquist's procedure, for finding a contiguous envy-free allocation for three players, and Dubins-Spanier, for computing a proportional allocation for any number of players. Dubins-Spanier is the only known moving knife procedure that can be simulated exactly in the RW model.

## 4 Envy-Free Allocations

Exact envy-free allocations are guaranteed to exist (see, e.g., Stromquist [63] and Su [65]), but cannot be computed by finite RW protocols for $n \geq 3$ players (see Stromquist [64]). For $n = 2$ players, an exact envy-free allocation can be computed with $O(1)$ queries via the Cut and Choose protocol.

An $\epsilon$-envy-free allocation with contiguous pieces can be found with $O(n/\epsilon)$ queries as follows.

**Theorem 4.1.** *For each cake cutting problem with $n \geq 3$ players and every $\epsilon > 0$, an $\epsilon$-envy-free allocation with contiguous pieces can be computed with $O(n/\epsilon)$ queries.*

*Proof sketch.* Let $K = \lceil 2/\epsilon \rceil$. Discretize the cake, by asking each player to divide the cake in many small cells ($K$ of them) of equal value $1/K$. For each player $i$, let $\widetilde{v}_i$ be the value density consistent with player $i$'s report and furthermore constant inside each interval generated by player $i$. Find an exact envy-free allocation $\widetilde{A}$ with contiguous pieces with respect to valuations $\widetilde{v} = (\widetilde{v}_1, \ldots, \widetilde{v}_n)$, which is guaranteed to exist. Then round each demarcation point of allocation $\widetilde{A}$ to the closest cut point reported by some player. The resulting allocation, $A$, is $\epsilon$-envy-free with respect to the true value densities $\vec{v}$. The total number of queries is $nK = O(n/\epsilon)$. $\square$

We further show that for three players, an $\epsilon$-envy-free allocation with contiguous pieces can be computed using only $O\left(\log \frac{1}{\epsilon}\right)$ queries.

**Theorem 4.2.** *For each cake cutting problem with $n = 3$ players and every $\epsilon > 0$, an $\epsilon$-envy-free allocation with contiguous pieces can be computed with $O\left(\log \frac{1}{\epsilon}\right)$ queries.*

---

[3]Ties are resolved deterministically, using for example the leftmost point with this property.

The idea is to approximately simulate a moving knife procedure due to Barbanel-Brams [7]. A detail that must be handled is that a discrete RW protocol does not allow the mediator to cut the cake directly, so the simulation has to search for the position of the "sword" in the moving knife protocol by keeping track of a small interval that must be made very small from the perspective of each player. The details can be found in the full version [20], together with the other omitted proofs.

Next we give a lower bound for three players; this will be extended to $n \geq 3$ players in Theorem 4.4.

**Theorem 4.3.** *For each $\epsilon > 0$, computing a contiguous $\epsilon$-envy-free allocation for three players requires $\Omega\left(\log \frac{1}{\epsilon}\right)$ queries.*

Generally there are many envy-free allocations, so the proof must ensure that for a long enough time none of these solutions are near. We will start from a family of valuations known as "rigid measure systems" [64], which was invented by Stromquist to show that no exact contiguous envy-free allocation can be computed in general in the RW model.

The difficulty is that while in [64] it was sufficient to avoid a single point in order for the protocol to not be able to find an exact envy-free solution with one more query, here we must avoid an entire interval and fit the valuations accordingly in order for an approximately envy-free solution to remain far away after each additional query. We first slightly generalize rigid measure systems from [64], to allow a different parameter $t_i$ for each player $i$. Then we will make a self reducible structure using this family of valuations, which will then be used to give the required lower bound.

**Definition 2.** *A tuple of value densities $(v_1, \ldots, v_n)$ is a generalized rigid measure system if:*

- *the density of each player $i$ is bounded by: $1/\sqrt{2} < v_i(x) < \sqrt{2}$, $\forall x \in [0, 1]$.*

- *there exist points $0 = x_0 < x_1 < \ldots < x_{n-1} < x_n = 1$ and values $s_i, t_i$ for each player $i$ such that $0 < s_i < 1/n < t_i < 1/2$ and $V_j(x_{j-1}, x_j) = V_j(x_j, x_{j+1}) = t_j$ for all $j = 1 \ldots n-1$ and $V_n(x_{n-1}, x_n) = V_n(0, x_1) = t_n$.*

See the full version [20] for an example. Generalized rigid measure systems satisfy the property that the valuations of the players for any given piece cannot differ too much.

**Lemma 1.** *Consider any cake cutting problem where for two players $i$ and $j$ where there exist $a, b > 0$ such that for all $x \in [0, 1]$, $1/a < v_i(x) < b$ and $1/a < v_j(x) < b$. Then for any two pieces $S_1, S_2$ of the cake, if $V_i(S_1) \geq ab \cdot V_i(S_2)$, it follows that $ab \cdot V_j(S_1) > V_j(S_2)$.*

A useful notion to measure how close a protocol is to discovering an approximately envy-free solution on a given instance will be that of a partial rigid measure system.

**Definition 3.** *A tuple of value densities $(v_1, v_2, v_3)$ is a partial rigid measure system if*

- *the density of each player $i$ is bounded everywhere by: $1/\sqrt{2} < v_i(x) < \sqrt{2}$, for all $x \in [0, 1]$.*

- *there exist values $k > 0$ and $1/2 > \ell_i > 1/3 > m_i > 0$ for each player $i$, and points $x, x', y, y' \in [0, 1]$, so that the matrix of valuations for pieces demarcated by these points is:*

|       | $[0, x]$ | $[x, x']$ | $[x', y]$ | $[y, y']$ | $[y', 1]$ |
|-------|----------|-----------|-----------|-----------|-----------|
| $V_1$ | $\ell_1$ | $k$       | $\ell_1$  | $k$       | $m_1$     |
| $V_2$ | $m_2$    | $k$       | $\ell_2$  | $k$       | $\ell_2$  |
| $V_3$ | $\ell_3$ | $k$       | $m_3$     | $k$       | $\ell_3$  |

Partial rigid measure systems have the next property: if $\mathcal{P} = \{z_1, \ldots, z_k\} \subset [0, 1]$ is a collection of cut points such that no points from $\mathcal{P}$ can be found in the intervals $(x, x')$ and $(y, y')$, then every contiguous partition attainable using only cut points from $\mathcal{P}$ has envy at least $0.01k$.

**Lemma 2.** *Let $(v_1, v_2, v_3)$ be a partial rigid measure system with parameters $k, m_i, \ell_i, x, x', y, y'$ so that $V_i([x, x']) = V_i([y, y']) = k$ $\forall i$. If $\mathcal{P} = \{z_1, \ldots, z_t\} \subset [0, 1]$ is a collection of cut points such that $(i)$ $x, x', y, y' \in \mathcal{P}$ and $(ii)$ no points from $\mathcal{P}$ can be found in the intervals $(x, x')$ and $(y, y')$, then each allocation with contiguous pieces demarcated by points in $\mathcal{P}$ has envy at least $0.01k$.*

Next we show how queries can be answered one at a time so that the valuations remain consistent with (some) partial rigid measure system throughout the execution of a protocol.

**Lemma 3.** *Suppose that at some point during the execution of an RW protocol for three players the valuations and cut points are consistent with a partial rigid measure system with parameters*

| | $[0, x]$ | $[x, x+k]$ | $[x+k, y]$ | $[y, y+k]$ | $[y+k, 1]$ |
|---|---|---|---|---|---|
| $V_1$ | $\ell_1$ | $k$ | $\ell_1$ | $k$ | $m_1$ |
| $V_2$ | $m_2$ | $k$ | $\ell_2$ | $k$ | $\ell_2$ |
| $V_3$ | $\ell_3$ | $k$ | $m_3$ | $k$ | $\ell_3$ |

Figure 2: Construction for envy-freeness.

*$k > 0$, $0 < m_i < 1/3 < \ell_i < 1/2$ for each $i$, and points $x, y$ (i.e. as in Figure 2). If the intervals $I = [x, x + k]$ and $J = [y, y + k]$ have no cut points inside, then a new cut query can be answered so that the valuations remain consistent with a partial rigid measure system where two new intervals $I' \subseteq I$ and $J' \subseteq J$ have no length $0.01k$ and no cut points inside, and the densities of all the players are uniform on $I'$ and $J'$.*

*Proof.* (of Theorem 4.3) Set the initial configuration to a partial rigid measure system with $k = 0.01$, $\ell_i = 0.35$, $m_i = 0.28$ for each player $i$. Let the initial cut points be $0.34, 0.35, 0.67, 0.68$ and set the intervals $I = [0.34, 0.35]$ and $J = [0.67, 0.68]$. It can be verified there exist compatible valuations for which the densities are in $(1/\sqrt{2}, \sqrt{2})$.

By iteratively applying Lemma 3 with every Cut query, the valuations remain consistent with a partial rigid system, where the intervals $I$ and $J$ always have uniform density, and their length cannot be diminished by a factor larger than 100 in each iteration. By Lemma 1, every configuration attainable with the existing cut points when given a partial rigid system as input has envy of at least $0.01k$. To get $\epsilon$-envy, we need $k/100 < \epsilon$, and so the number of queries is $\Omega\left(\log \epsilon^{-1}\right)$. □

The construction can be extended to give a lower bound for any number of players.

**Theorem 4.4.** *Let $n \geq 3$ be fixed. For each $\epsilon > 0$, computing an $\epsilon$-envy-free allocation with contiguous pieces for $n \geq 3$ players requires $\Omega\left(\log \frac{1}{\epsilon}\right)$ queries.*

The lower bound of $\Omega\left(\log \frac{1}{\epsilon}\right)$ is in fact tight for the class of generalized rigid measure systems, for any fixed number of players. To show this, we prove an upper bound of $O\left(\log \frac{1}{\epsilon}\right)$ for this class by designing a moving knife procedure and then simulating it in the RW model.

**Theorem 4.5.** *For the class of generalized rigid measure systems, an $\epsilon$-envy-free allocation with contiguous pieces can be computed with $O\left(\log \frac{1}{\epsilon}\right)$ queries for every fixed number $n$ of players.*

## 5 Perfect Allocations

We show that for $n = 2$ players, the problem of computing $\epsilon$-perfect allocations can be solved more efficiently by simulating Austin's moving procedure in the RW model.

**Theorem 5.1.** *For each cake cutting problem with $n = 2$ players and every $\epsilon > 0$, an $\epsilon$-perfect allocation can be computed with $O(\log \frac{1}{\epsilon})$ queries.*

We also show this bound is optimal by giving a matching lower bound.

**Theorem 5.2.** *For each $\epsilon > 0$, computing an $\epsilon$-perfect allocation with the minimum number of cuts for $n = 2$ players requires $\Omega\left(\log \frac{1}{\epsilon}\right)$ queries.*

We prove the lower bound by maintaining throughout the execution of any protocol two intervals in which the cuts of the perfect allocation must be situated, such that the distance to a perfect partition cannot decrease too much with any cut query. The work in [21] gave a (weaker) lower bound for finding $\epsilon$-perfect allocations between two players in the communication model, which implies a lower bound for query complexity (in the RW model or any other query model).

**Lemma 4.** *Consider a cake cutting instance with $n = 2$ players and let $\epsilon > 0$. Suppose a protocol made a sequence of queries such that the valuations are consistent with Figure 3, for some parameters $x, y, a, b, c, d, e > 0$, such that player 1 has uniform density everywhere, $y = x + 0.5$, $0 < a, d \leq 0.1$, $x, b, c, e > 0$, $x + a + b = 0.5$ $c + 2d + e = 0.5$, and moreover:*

    *1. $\epsilon < 0.001 \min\{a, d\}$*

| | $[0,x]$ | $[x,x+a]$ | $[x+a,y]$ | $[y,y+a]$ | $[y+a,1]$ |
|---|---|---|---|---|---|
| $V_1$ | $x$ | $a$ | $0.5-a$ | $a$ | $b$ |
| $V_2$ | $c$ | $d$ | $0.5-2d$ | $3d$ | $e$ |

Figure 3: Construction for the perfect lower bound.

2. *every allocation demarcated by cuts $k,\ell \in (0,1)$ with $k < \ell$ that is $\epsilon$-perfect from the point of view of player 1 is worth to player 2 less than $0.5 - d/100 - \epsilon$ when $k < x$ and more than $0.5 + d/100 + \epsilon$ when $k > x + a$.*

3. *there are no cut points inside the intervals $I = [x, x+a]$ and $J = [y, y+a]$.*

*Then one more query can be answered so that the valuations remain consistent with the history of the protocol and conditions 2 and 3 still hold with respect to new parameters $x', a' = a/100, d' = d/100$ and intervals $I' = [x', x' + a'], J' = [y', y' + a']$.*

*Proof of Theorem 5.2.* Set the initial configuration consistent with Figure 3 for a suitable initial choice of parameters. Then, by iteratively applying Lemma 4 with every cut query received, we will obtain that the number of queries is $\Omega\left(\log \frac{1}{\epsilon}\right)$. $\qquad \square$

## 6 Equitable Allocations

Cechlarova, Dobos, and Pillarova [24] showed that for any number of players and any order, there exists a contiguous equitable allocation in that order. Moreover, the equitable allocation is proportional for some order of the players.

**Theorem 6.1** ( Cechlarova and Pillarova [25]). *For every fixed number $n$ of players and each $\epsilon > 0$, a contiguous $\epsilon$-equitable and proportional allocation can be computed with $O(\log \frac{1}{\epsilon})$ queries.*

We give a matching lower bound for the number of queries required for finding $\epsilon$-equitable allocations with contiguous pieces between two players.

**Theorem 6.2.** *For each $\epsilon > 0$, computing an $\epsilon$-equitable allocation with contiguous pieces for two players requires $\Omega\left(\log \frac{1}{\epsilon}\right)$ queries.*

For two hungry players, the contiguous equitable and proportional allocation is unique.

**Lemma 5.** *For two players with hungry valuations, the cut point of the equitable allocation is unique.*

Next we show that when the valuations of the players are as in the next figure and no cuts may be used from the interval $(x, y)$, the distance from a contiguous equitable allocation is high.

| | $[0,x]$ | $[x,y]$ | $[y,1]$ |
|---|---|---|---|
| $V_1$ | $0.5+a$ | $b-a$ | $0.5-b$ |
| $V_2$ | $0.5-b$ | $b-a$ | $0.5+a$ |

Figure 4: Construction for equitable lower bound. The distance from a contiguous equitable and proportional allocation is $b - a$, where $0 < a < b < 0.5$ and $0 < x < y < 1$.

**Lemma 6.** *Consider a two player problem where there exist points $0 < x < y < 1$ and values $0 < a < b < 0.5$ such that the valuations are consistent with Figure 4, where $V_1(0, x) = 0.5 + a = V_2(y, 1)$, $V_2(x, 1) = 0.5 + b = V_1(0, y)$. Then every contiguous allocation that uses cut points outside the interval $(x, y)$ has distance at least $b - a$ from equitability.*

Given such a configuration, queries can be handled in a way that preserves the symmetry and the distance to equitability gets reduced by a constant factor.

**Lemma 7.** *Consider a two player problem where there exist points $0 < x < y < 1$ and values $0 < a < b < 0.5$ such that $V_1(0, x) = 0.5 + a = V_2(y, 1)$, $V_2(x, 1) = 0.5 + b = V_1(0, y)$. Then any Cut query (addressed to either player) can be answered so that the new configuration has two new points $z < t$ such that $z, t \in (x, y)$, the valuations satisfy $V_1(0, z) = 0.5 + a' = V_2(t, 1)$, $V_2(z, 1) = 0.5 + b' = V_1(0, t)$, and $b' - a' \geq (b - a)/100$.*

The proof of Theorem 6.2 follows by combining the previous lemmas (see full version [20]).

## 7  Moving Knife Protocols

We will consider a family of protocols that seems to, on one hand, capture all types of protocols that have so far been called "moving knife" procedures and, on the other hand, be simple enough for a transparent simulation. An important ingredient of the definition is that knife positions must be continuous. To ensure that "cut queries" fall within the definition, we will only require continuity for hungry valuation functions. The formal definition can be found in the full version of the paper [20].

**Theorem 7.1.** *Consider a cake cutting problem where the value densities are bounded from above and below by strictly positive constants. Let $\mathcal{M}$ be an RW moving knife protocol with at most $r$ steps, such that $\mathcal{M}$ outputs $\mathcal{F}$-fair allocations demarcated by at most a constant number $C$ of cuts.*

*Then for each $\epsilon > 0$, there is an RW protocol $\mathcal{M}_\epsilon$ that uses $O\left(\log \frac{1}{\epsilon}\right)$ queries and computes $\epsilon$-$\mathcal{F}$-fair partitions demarcated with at most $C$ cuts.*

**Theorem 7.2.** *The Austin, Austin's extension, Barbanel-Brams, Stromquist, Webb, Brams-Taylor-Zwicker, and Saberi-Wang moving knife procedures can be simulated with $O\left(\log \frac{1}{\epsilon}\right)$ RW queries when the value densities are bounded from above and below by positive constants.*

**An Equitable Protocol**: Next we show a simple moving knife protocol in the Robertson-Webb model for computing equitable allocations for any number of hungry players. A moving knife procedure for computing exact equitable allocations that works even when the valuations are not hungry was discovered independently by Segal-Halevi [60].

***Equitable Protocol*** : *Player $1$ slides a knife continuously across the cake, from $0$ to $1$. For each position $x_1$ of the knife, player $1$ is asked for its value of the piece $[0, x_1]$; then each player $i = 2 \ldots n$ iteratively positions its own knife at a point $x_i \in [x_{i-1}, 1]$ with $V_i(x_{i-1}, x_i) = V_1(0, x_1)$ if possible, and at $x_i = 1$ otherwise.*

*Player $n$ shouts "Stop!" when its own knife reaches the right endpoint of the cake (i.e., $x_n = 1$). The cake is allocated in the order $1 \ldots n$, with cuts at $x_1 \ldots x_{n-1}$.*

**Theorem 7.3.** *There is an RW moving knife protocol that computes a contiguous equitable allocation for every cake cutting instance with $n$ hungry players.*

This immediately implies a moving knife protocol for computing an allocation that is not only equitable, but also proportional; this can be achieved by running Equitable Protocol for every permutation of the players and choosing the one that is proportional.

## 8  The Stronger and Weaker Models

We also discuss two other query models. The first one, which we call $RW^+$, is stronger in that the inputs to evaluate queries need not be previous cut points and at the end the protocol can use arbitrary points (i.e. not just cuts discovered through queries) to demarcate the final allocation.

**Definition 4** ($RW^+$ query model). *An $RW^+$ protocol for cake cutting communicates with the players via two types of queries:*

- **Cut$_i(\alpha)$**: *Player $i$ cuts the cake at a point $y$ where $V_i([0, y]) = \alpha$, for any $\alpha \in [0, 1]$.*

- **Eval$_i(y)$**: *Player $i$ returns $V_i([0, y])$, for any $y \in [0, 1]$.*

*At the end of execution an $RW^+$ protocol outputs an allocation that can be demarcated by any points of its choice, regardless of whether they have been discovered through queries or not.*

The $RW^+$ model differs from the $RW$ model in subtle ways. For instance, in $RW$ there exists a characterization of truthful protocols (i.e. all truthful protocols are dictatorships for $n = 2$ players, with a similar statement for $n \geq 3$ players [19]). A similar characterization is not known in the $RW^+$ model. However the $RW^+$ model allows a general simulation of moving knife protocols without requiring that valuations are bounded from below. Our constructions for the lower bounds work against this stronger query model.

The lower bounds from the RW model still hold, since our constructions did not use in any crucial way the fact that the evaluate inputs must come from previous cut queries.

**Corollary 1.** *Computing an $\epsilon$-envy-free allocation with contiguous pieces among $n = 3$ players, an $\epsilon$-perfect allocation with two cuts between $n = 2$ players, and an $\epsilon$-equitable allocation with contiguous pieces between $n = 2$ players in the $RW^+$ models requires $\Theta(\log \frac{1}{\epsilon})$ queries. Computing a contiguous $\epsilon$-envy-free allocation for any fixed number $n$ of players requires $\Omega(\log \frac{1}{\epsilon})$ queries.*

In the $RW^+$ model we can simulate moving knife protocols without the requirement that the valuations are bounded from below since the center can reduce (half) the time directly with each iteration, instead of reducing it through the lens of the players' valuations.

**Theorem 8.1.** *Consider a cake cutting problem where the value densities are bounded from above by constant $D > 0$. Let $\mathcal{M}$ be an $RW^+$ moving knife protocol with at most $r$ steps, such that $\mathcal{M}$ outputs $\mathcal{F}$-fair allocations demarcated by at most a constant number $C$ of cuts.*

*Then for each $\epsilon > 0$, there is an $RW^+$ protocol $\mathcal{M}_\epsilon$ that uses $O\left(\log \frac{1}{\epsilon}\right)$ queries and computes $\epsilon$-$\mathcal{F}$-fair partitions demarcated with at most $C$ cuts.*

We also introduce a weaker model, which we call $RW^-$, where the protocol can ask the players only the evaluate type of query.

**Definition 5** ($RW^-$ query model)**.** *An $RW^-$ protocol for cake cutting communicates with the players via queries of the form*

- **Eval**$_i(y)$*: Player $i$ returns $V_i([0, y])$, where $y \in [0, 1]$ is arbitrarily chosen by the center.*

*At the end an $RW^-$ protocol outputs an allocation that can be demarcated by any points.*

If the valuations are arbitrary, then an $RW^-$ protocol may be unable to find any fair allocation at all. The reason is that no matter what queries an $RW^-$ protocol asks, one can hide the entire instance in a small interval that has value 1 for all the players; this interval will shrink as more queries are issued, but can be set to remain of non-zero length until the end of execution.

However, if the valuations are bounded from above[4], then an $RW^-$ protocol is quite powerful.

**Theorem 8.2.** *Suppose the valuations of the players are bounded from above by a constant $D > 0$. Then any $RW^+$ query can be answered within $\epsilon$-error using $O(\log \frac{1}{\epsilon})$ $RW^-$ queries.*

*Proof.* Let there be an instance with arbitrary valuations $v_1 \ldots v_n$ such that $v_i(x) < D$ for all $x \in [0, 1]$ and $i \in N$. Since an $RW^-$ protocol can use the same type of evaluate queries as an $RW^+$ protocol, the simulation has to handle the case where the incoming query is a cut. Let this be $Cut_i(\alpha)$ and denote by $x$ the correct answer to the query. In order to find an approximate answer using only evaluate queries, initialize $\ell = 0$, $r = 1$, and search for the correct answer: ($*$) Let $m = (\ell + r)/2$. Ask player $i$ the query $Eval_i(m)$ and let $w$ be the answer given. If $|w - \alpha| \le \epsilon$, return $m$. Otherwise, if $m > \alpha$, set $r = m$, and if $m < \alpha$, set $\ell = m$; return to ($*$). This procedure halves the interval $[\ell, r]$ with every iteration. Moreover, from the bound on the valuations, an interval of length $\epsilon/D$ cannot be worth more than $\epsilon$ to any player. Thus the search stops in $O(\log \epsilon^{-1})$ rounds. $\qquad\square$

## 9 Discussion

An important open question is to obtain stronger lower bounds for $n \ge 4$ players for computing contiguous envy-free allocations and for $n \ge 3$ players for perfect allocations with minimal number of cuts. We conjecture that unlike equitability, which remains logarithmic in $1/\epsilon$ for any number of players, computing a contiguous $\epsilon$-envy-free allocation for $n = 4$ players and an $\epsilon$-perfect allocation with minimal cuts for $n = 3$ players will require $\Omega(\frac{1}{\epsilon})$ queries. Since moving knife protocols can be simulated with $O(\log \frac{1}{\epsilon})$ queries, this would imply that no moving knife protocol exists for computing an envy-free allocation for $n \ge 4$ players or a perfect allocation for $n \ge 3$ players (the existence of such procedures has been posed as an open question, e.g. in [12]).

**Acknowledgements**   We thank the reviewers for helpful feedback. This project has received funding from the European Research Council (ERC) under the European Union's Horizon 2020 Research and Innovation Programme (grant agreement no. 740282). A part of this work was done while Simina Brânzei was visiting the Simons Institute for the Theory of Computing.

---

[4]There exist other types of valuations on which the $RW^-$ model may be useful, such as piecewise constant valuations defined on a grid, with the demarcations between intervals of different height known to the protocol.

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
