# OpenReview forum: "The Query Complexity of Cake Cutting"
_NeurIPS.cc/2022/Conference — NeurIPS 2022 Accept_

### Official Review · Reviewer_XBqt · 2022-06-29

**Rating:** 8
**Confidence:** 4
**Soundness:** 3 good
**Presentation:** 3 good
**Contribution:** 4 excellent

**Summary:**


The paper is in two parts
I) The authors prove new upper and lower bounds in the RW model of cake cutting, which is a model for
discrete protocols. The main operations are CUT and QUERY. In all problems each player has a valuation
function on pieces of cake that may differ quite a lot from the other players. The authors consider
three types of cake cutting. In all of them assume that there are n people A1,...,An and that at the
end Ai gets Pi. They consider three types of outcomes

a) epsilon-envy free. For all i, Ai thinks that Pi is \ge Aj +\epsilon AND Pi is connected. They show

--- for n=3 there is an upper bound of O(log (1/epsilon) queries and a matching lower bound of
Omega(1/epsilon) queries.

--- for n\ge 4 an upper bound of O(n/epsilon) queries and a lower bound of Omega(1/epsilon) queries

b) epsilon-perfect. For all i,j Ai thinks Pj is size 1/n PM epsilon. Assume 2 players only.

-- if the min number of cuts is used then this can be done with O(1/epsilon) queries

-- if 2 cuts are used then this requires Omega(1/epsilon).

c) epsilon-equitable if there is some c such that, for all i, Ai thinks Pi is c PM epsilon.
For 3 players AND the pieces are connected, this requires Omega(\log 1/epsilon).

2) There is another cake cutting paradigm: Moving Knife. This paper introduces a model
for MK protocols that captures all current MK protocols. They show that all moving knife
protocols can be simulated by O(log(1/epsilon) queries in the RW model. This allows one to easily get
lower bounds on moving knife protocols. These lower bounds match the upper bounds.



**Questions:**


1) What does your model of Moving Knife protocols say about the
non-approximate moving knife protocols. The  epsilon=0 case.

2) The proofs look correct but I could not tell what the new ideas were.




**Limitations:**


This paper has no societal impact.
It is about a very interesting math problem.


**Strengths And Weaknesses:**

1) The envy-free protocols have always had the problem of the pieces being very disconnected.
This paper addresses that by looking at epsilon-envy free for which they have protocol
and lower bounds. For n=3 the upper and lower bounds match.

2) The moving knife protocols have not had a good model or any lower bounds. This paper
provides both.

These results are original, significant, and clear.

Weaknesses:
1) There seem to be two parameters: number of queries and number of cuts. The paper does
not mention the number-of-cuts for many of their algorithms, making me wonder if they are
galactic.

This is unclear.

---

> ### Author Response · Authors · 2022-08-02
> **Response to reviewer XBqt**
>
> Thank you for your review.
>
> ### 1. Regarding the number of cuts versus queries:
> + The number of queries is an upper bound on the number of cuts.  For example, we show how an epsilon-envy-free allocation with contiguous pieces can be obtained by asking log(1/epsilon) queries (so in particular asking to make at most log(1/epsilon) cut queries and at most log(1/epsilon) eval queries). However, the final allocation will give a contiguous interval to each player.
>
> ### 2. What does your model of Moving Knife protocols say about the non-approximate moving knife protocols? The epsilon=0 case:
> + The epsilon = 0 case corresponds to finding exact solutions (e.g. exact envy-free allocations).
> + Lower bounds of the form log(1/epsilon) or 1/epsilon for moving knife protocols imply that in the epsilon = 0 case, infinitely many RW operations are needed to find the desired allocation with no error.
> + Upper bounds show how such allocations can be found by an RW protocol in the limit.
>
> ### 3. The proofs look correct but I could not tell what the new ideas were.
> + We will comment on this. Briefly, for the problems we consider (e.g. envy-free, perfect, or equitable), a solution is guaranteed to exist. Thus an adversary can never be so adversarial as to completely throw away the solutions in the process of answering queries. The challenge then is how to keep hiding the solutions. The method we use hides the solution in a sort of self-reducible way, where the structure looks the same each time but with changing  parameters.

---

> > ### Comment · Reviewer_XBqt · 2022-08-07
> > **Response to Response to reviewer XBqt (me)**
> >
> > The authors have satisfied the few negative comments I had.
> > I originally gave the paper an 8-strong accept, and I still give it that rating.

---

### Official Review · Reviewer_NhHm · 2022-07-05

**Rating:** 4
**Confidence:** 4
**Soundness:** 4 excellent
**Presentation:** 2 fair
**Contribution:** 3 good

**Summary:**

The paper does a nice job expositing and motivating the question of query complexity of \eps-EF cake cutting. Unfortunately, it seems quite outdated, missing important references from the past ~5 years. (I usually wouldn't be too worried about missing references, but the technical contributions in the paper are not so impressive so I view the main contribution as motivating the questions, but then the context is more important.)

I’m also not sure that this is a good fit for NeurIPS, even interpreting the scope quiet liberally.

UPDATE: Given the authors' responses I went back and re-read the paper.
I think it's nicer than I had originally assessed and will increase the score.
However, I'm still skeptical about whether this is a great fit for NeurIPS.

As a minor note, I also realized that the $\Omega(\log(1/\epsilon))$ lower bound only holds against deterministic protocols.


**Questions:**

UPDATE: These questions have been answered by reviewers.

How does your submission relate to a paper by Brânzei and Nisan (EC’19) on communication complexity of cake cutting?


How does your protocol for 3 agents compare to the protocol by Deng et al for monotone valuations?


**Strengths And Weaknesses:**

See summary above.

---

> ### Author Response · Authors · 2022-08-02
> **Response to reviewer NhHm**
>
> Thank you for your review.
>
> ### 1. How does your submission relate to a paper by Brânzei and Nisan (EC’19) on communication complexity of cake cutting?
> + The communication complexity bounds are weaker than the ones we give for query complexity, since it is even more challenging to prove lower bounds in the communication model. Also, Brânzei and Nisan (EC’19) do not give a lower bound for epsilon envy-freeness.
>
> ### 2. How does your protocol for 3 agents compare to the protocol by Deng et al for monotone valuations?
> + The paper by Deng et al studies a class of valuations where the *utility* of a player does not depend only on what they get, but may depend on the allocations of others. This means that their lower bounds do not apply and the exponential upper bounds they give are not tight in our setting. Most of the literature dating back to Steinhaus has focused on additive valuations, which we study; these are relevant in many other resource allocation problems (e.g. auctions).
>
> ### 3. "I’m also not sure that this is a good fit for NeurIPS, even interpreting the scope quiet liberally":
> + Algorithmic game theory is explicitly listed in the call for papers. We also believe the conversation of how to design fair algorithms in society can gain from different perspectives on fair resource allocation.
>
> ### 4. Related literature:
> + To the best of our knowledge, there has been no progress on these bounds since our work (by us or others, we have tried to improve them but it seems additional ideas are needed). We will add a paragraph to discuss more papers published in the last few years. We wanted to add this with the rebuttal but were concerned that doing so may violate the anonymity rules. If the reviewer has specific suggestions for what papers to cite we would be happy to include them and we apologize for missing anything.

---

> > ### Comment · Reviewer_NhHm · 2022-08-03
> > **I'm still confused about relation to prior work**
> >
> > 1. Since communication complexity is a stronger model, do their lower bounds apply to your setting?
> > Shouldn't you at least cite this paper and discuss it in more detail?
> >
> > 2. For $n=3$ agents, Deng et al give an efficient algorithm (not hardness) for _monotone valuations_ (not general "utilities"). How does this result compare to your efficient protocol for n=3 agents?

---

> > > ### Author Response · Authors · 2022-08-04
> > > **Re: I'm still confused about relation to prior work**
> > >
> > > 1. Regarding communication versus query complexity:
> > > Yes, we will clarify this in the additional paragraph of related work we promised to add. We actually thought we had cited Branzei and Nisan '19 but forgot. Thank you for checking this point with us.
> > > To summarize, the differences are:
> > > - Branzei and Nisan '19 don't give any lower bounds for envy-freeness.
> > > - the lower bounds for equitable and perfect allocations in Branzei and Nisan '19 are weaker than the ones we show (which are tight), since the communication model is more challenging to prove bounds in.
> > >
> > > 2. Regarding Deng et al:
> > > + For the class of monotone valuations, the bound for n=3 players in Deng et al is Theta(log^2(K/epsilon)), where K is the Lipschitz constant of the valuations. In contrast, in the additive model we focus on---where the valuations are induced by probability measures---the bound for three players is Theta(log(1/epsilon)), so it has a different power and no dependence on a Lipschitz constant.
> > > + As a side note, for the white box complexity, where the utilities are given as circuits,  Deng et al show (in the more general model of utilities they consider), that the problem of finding an envy-free allocation with contiguous pieces is PPAD-complete. The problem is considerably harder for additive valuations and in fact it's still open.

---

> > > > ### Author Response · Authors · 2022-08-04
> > > > **Re:**
> > > >
> > > > Just to clarify, in the last phrase about the white box complexity with additive valuations: the problem is harder---meaning "more challenging"---with additive valuations compared to more general ones.

---

> > > ### Author Response · Authors · 2022-08-04
> > > **An important clarification**
> > >
> > > The paper by Branzei and Nisan '19 cites our work and builds on it (e.g. uses our definition of moving knife protocols). We agree we should cite them too and we will.

---

### Official Review · Reviewer_X5uM · 2022-07-11

**Rating:** 6
**Confidence:** 4
**Soundness:** 4 excellent
**Presentation:** 4 excellent
**Contribution:** 3 good

**Summary:**

The paper studies various problems in the fair cake cutting area, and advances theoretical state of the art by providing improved lower and upper bounds.

In particular, three notions of fairness are considered: envy-freeness, proportionality, and equitability (plus the combinations of the three, called "perfect"). Because of strong impossibility results, the authors focus on the approximate variants of these fairness concepts.

They establish a lower bound of $O(\log 1/\epsilon)$ for the query complexity of three problems: connected $\epsilon$-envy-free (for three players), connected $\epsilon$-equitable and $\epsilon$-perfect allocation with two cuts (for two players). They also present almost tight upper bounds for the 1st & 3rd problems (the other ones already had an upper bound matching the new lower bound).

While essentially nothing was known for $\epsilon$-envy-free, the other two notions already had _similar_ upper bounds for 3 (as opposed to 2) players, with upper bounds that still are pretty loose.

They show how a large class of cake-cutting methods can be simulated using Robertson-Webb query model (where an outsider asks players about their preferences until a solution can be proposed).

**Questions:**

none

**Strengths And Weaknesses:**

Strengths:
- Advances state of the art and closes the gap for certain settings.

Weaknesses:
- The results and methods seem incremental.
- Relevance to the community is unclear.
- No real application discussed, hence no experiments.

---

> ### Author Response · Authors · 2022-08-02
> **Response to reviewer X5uM**
>
> Thank you for your review.
>
> ### 1. Results and methods:
> + The method of hiding the solution using the type of self-reducible structure that we use (e.g. for the envy-free and perfect lower bounds) may be more widely applicable. For higher number of players, we expect that additional ideas will be needed. It has often been the case that problems with a few players were solved before successful generalizations to more players were built.
>
> ### 2. Relevance to community:
> + Fairness is an important topic; we believe that the understanding of how to design fair algorithms can improve when considering different perspectives on fair resource allocation (from various settings and strands of literature).

---

### Official Review · Reviewer_KLGZ · 2022-07-11

**Rating:** 6
**Confidence:** 2
**Soundness:** 3 good
**Presentation:** 3 good
**Contribution:** 3 good

**Summary:**

This paper considers several approximate fairness allocation problems among n players (envy-free, perfect, and equitable), and considers query complexity in the Robertson-Webb model (RW). The authors show lower bounds and upper bounds for approximate envy-free allocation (no player prefers another player's piece to her own) for any number of players. For approximate perfect and equitable allocations, improvements were made to the case n=2 players. To achieve their upper bounds, the authors formalize "moving-knife protocols" (MKPs) within the RW model.

**Questions:**


No questions.

**Limitations:**


Yes.

**Strengths And Weaknesses:**

S1. Novel theoretical upper and lower bounds obtained for approximate allocation.

S2. For epsilon-envy-free allocation, algorithms are efficient, i.e. O( log 1/epsilon ) for 3 players, O( n / epsilon ) for n players. Also, the upper bound in 3-player case matches the lower bound proven.

S3. It was unknown if MKPs could be formalized within the RW model. The fact that they can is quite interesting, in my opinion. This unifies two seemingly disjoint techniques.

W1. With the exception of approximate envy-free allocation, the method of authors did not improve existing upper or lower bounds for n >= 3 players, only for n = 2.

W2. Lower bound for approximate envy-free allocation is not tight for n >= 4 players.

---

> ### Author Response · Authors · 2022-08-02
> **Response to reviewer KLGZ**
>
> Thank you for your review.

---

### Meta-Review · Area_Chair_7E27 · 2022-08-27

**Recommendation:** Accept
**Confidence:** Certain

**Metareview:**

This paper considers the well-studied cake-cutting problem that captures the fair division of a resource (e.g., time, mineral deposits, fossil fuels, and many others) among n parties with equal rights but different interests over the resource. This is also a natural problem in algorithmic game theory. Various such settings are considered where the i'th party gets part P(i) at the end of the protocol. Approximation (various notions of, parametrized by some error epsilon > 0) are essential to the results in the paper.

In the first part of the paper, new upper and lower bounds are developed in the Robertson-Webb model of discrete protocols. For instance, "epsilon-envy-free" protocols are considered (for all (i,j), player i thinks---according to their valuation function---that P(i) is at least P(j) - epsilon AND P(i) is connected) and it is shown, e.g., that: (a) for n=3, Theta(log (1/epsilon)) queries are necessary and sufficient; and (b) for n >= 4, an upper bound of O(n/epsilon) queries and a lower bound of Omega(1/epsilon) queries. This is just a sample. In the second part of the paper, another cake-cutting approach---Moving Knife (MK)---is considered: the paper introduces a model for MK protocols that captures all current MK protocols and shows new results.

This is a comprehensive paper on a natural mathematical model for fair division.

**Award:**

No

---

### Decision · Program_Chairs · 2022-09-14

Accept